# Chain-of-Thought Tuning: Masked Language Models can also Think Step By Step in Natural Language Understanding

**Caoyun Fan[1], Jidong Tian[1], Yitian Li[1], Wenqing Chen[2], Hao He[1†], Yaohui Jin[1†]**

[1] MoE Key Lab of Artificial Intelligence, AI Institute, Shanghai Jiao Tong University
{fcy3649, frank92, yitian_li, hehao, jinyh}@sjtu.edu.cn
[2] School of Software Engineering, Sun Yat-sen University
chenwq95@mail.sysu.edu.cn

## Abstract

Chain-of-Thought (CoT) is a technique that guides Large Language Models (LLMs) to decompose complex tasks into multi-step reasoning through intermediate steps in natural language form. Briefly, CoT enables LLMs to think step by step. However, although many Natural Language Understanding (NLU) tasks also require thinking step by step, LLMs perform less well than small-scale Masked Language Models (MLMs). To migrate CoT from LLMs to MLMs, we propose Chain-of-Thought Tuning (CoTT), a two-step reasoning framework based on prompt tuning, to implement step-by-step thinking for MLMs on NLU tasks. From the perspective of CoT, CoTT's two-step framework enables MLMs to implement task decomposition; CoTT's prompt tuning allows intermediate steps to be used in natural language form. Thereby, the success of CoT can be extended to NLU tasks through MLMs. To verify the effectiveness of CoTT, we conduct experiments on two NLU tasks: hierarchical classification and relation extraction, and the results show that CoTT outperforms baselines and achieves state-of-the-art performance.

## 1 Introduction

Chains-of-Thought (CoT) (Wei et al., 2022b; Fu et al., 2023; Zhang et al., 2022) is a technique to help language models think step by step (Kojima et al., 2022). Through intermediate steps in natural language form, CoT can guide language models to decompose complex tasks into multi-step reasoning processes. Currently, CoT is mainly employed in Large Language Models (LLMs) (Brown et al., 2020; Chowdhery et al., 2022; Touvron et al., 2023; Zeng et al., 2023; OpenAI, 2023), as LLMs demonstrate impressive complex reasoning capabilities (Wei et al., 2022a; Zhao et al., 2023).

However, although LLMs achieved state-of-the-art performance on a wide range of NLP tasks

---

†Corresponding author.

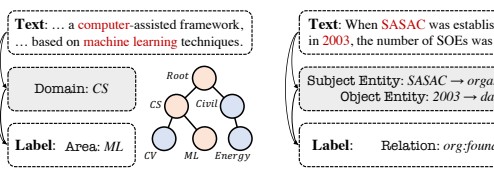

(a) Hierarchical Classification    (b) Relation Extraction

Figure 1: In NLU tasks, various forms of intermediate steps (drawn in gray) would exist as reasoning evidence between **Text** and **Label**.

(Zhao et al., 2023), Yang et al. (2023); Kocon et al. (2023); Lai et al. (2023) found that LLMs were less competitive than small-scale Masked Language Models (MLMs) (Devlin et al., 2019; Liu et al., 2019; Lan et al., 2020) in many traditional Natural Language Understanding (NLU) tasks (e.g. tasks in GLUE (Wang et al., 2019b) and SuperGLUE (Wang et al., 2019a)). The reasons for this are manifold: on the one hand, LLMs are typically autoregressive models and are not well-suited for NLU tasks (Liu et al., 2021; Raffel et al., 2020); on the other hand, NLU tasks usually involve rich domain knowledge (Fan et al., 2023a; Zhang et al., 2023) and require fine tuning to master such knowledge (Fan et al., 2023b,c; Wang et al., 2023). Therefore, we wonder whether the success of CoT in LLMs can be transferred to MLMs in NLU tasks.

In fact, many NLU tasks also require thinking step by step. For example, in hierarchical classification (Jr. and Freitas, 2011; Kowsari et al., 2017), each text should follow a pre-defined taxonomic hierarchy to generate multiple labels in turn, so that high-level labels can be considered as the intermediate step, as shown in Fig. 1(a); in relation extraction (Zhang et al., 2017; Alt et al., 2020; Stoica et al., 2021), the entity types of the subject and the object in each instance need to be determined in advance of annotating the relation (Zhou and Chen, 2021), so the entity types can be considered as the intermediate step, as shown in Fig. 1(b). Although

previous studies (Chen et al., 2020b; Han et al., 2021; Zhou and Chen, 2021) have also attempted to incorporate intermediate steps into language models, in the perspective of CoT, these methods lack both a decomposition process for tasks (not multi-step reasoning) and explicit use of intermediate steps (not in natural language form).

In this study, we propose Chain-of-Thought Tuning (CoTT), a two-step reasoning framework based on prompt tuning (Sun et al., 2022; Liu et al., 2022), to implement step-by-step thinking for MLMs on NLU tasks. The two steps of CoTT are:

> **Step I**: *MLM generates the intermediate step $\hat{I}$ based on the text $x$.*
> **Step II**: *MLM predicts the final result $y$ based on the text $x$ and the intermediate step $\hat{I}$.*

CoTT effectively addresses the shortcomings of previous methods: on the one hand, CoTT's two-step framework enables MLM to implement task decomposition; on the other hand, CoTT is based on prompt tuning, which allows intermediate steps to be used in natural language form (Schick and Schütze, 2021b; Gu et al., 2022). In order to inject/generate intermediate steps flexibly in both steps, we propose *convertible slot* [C], a new type of template slot (Petroni et al., 2019; Liu et al., 2022) in prompt tuning.

To evaluate the effectiveness of CoTT, we conduct extensive experiments on two traditional NLU tasks: hierarchical classification and relation extraction, and the experimental results reveal that CoTT obtains superior performance compared with baselines and achieves state-of-the-art performances. Furthermore, due to the introduction of intermediate steps, CoTT is no longer an end-to-end method but can display the reasoning process. Therefore, we can further improve the ability of CoTT by monitoring the reasoning process.

We summarize our contributions as follows:

- Based on the philosophy of CoT, we propose a two-step reasoning framework to enable MLMs to think step by step in NLU tasks, and we call it Chain-of-Thought Tuning (CoTT).

- We propose *convertible slot* [C], a new type of slot in prompt tuning, which can flexibly inject or generate intermediate steps depending on scenarios.

- We evaluate our CoTT on two NLU tasks: hierarchical classification and relation extrac-

tion, and the experimental results demonstrate the effectiveness of CoTT.

## 2 Related Work

### 2.1 Chain-of-Thought

Chain-of-Thought (CoT) (Wei et al., 2022b) is a prompt technique that elicits LLMs to produce intermediate reasoning steps leading to the final answer. Recent studies (Zhou et al., 2023; Fu et al., 2023; Zhang et al., 2022) have confirmed that CoT can substantially improve the reasoning ability of LLMs. Studies (Zhang et al., 2022) have shown that LLMs can perform CoT reasoning with zero-shot scenarios (Kojima et al., 2022) or manually written few-shot scenarios (Wei et al., 2022b). In zero-shot scenarios, Kojima et al. (2022) showed that LLMs can generate decent intermediate steps, by adding certain magic phrases like "Let's think step by step", Zelikman et al. (2022) employed LLM to generate many intermediate steps and those intermediate steps that can lead to the final answer are chosen. In few-shot scenarios, LLM's reasoning ability can be improved by a few effective demonstrations on multi-step reasoning tasks. Some studies discussed how to select demonstrations efficiently: Fu et al. (2023) considered prompts with higher reasoning complexity (more intermediate steps) to be efficient demonstrations, Rubin et al. (2022) automatically constructed demonstrations based on the semantic similarity of texts. However, it is still unknown whether CoT can be applied to small-scale language models.

### 2.2 Prompt Tuning

Since the advent of GPT-3 (Brown et al., 2020), prompt tuning (Sun et al., 2022; Liu et al., 2022) has received considerable attention. Prompt tuning (Schick et al., 2020; Gu et al., 2022) aims to transform the downstream tasks into the pre-training tasks of Pre-trained Language Models (PLMs) with appropriate manual prompts, which can bridge their gap and better utilize PLMs (Han et al., 2021; Chen et al., 2022a). Although the origin of prompt tuning is large-scale autoregressive language models, the following studies (Schick and Schütze, 2021a,b) found that small-scale Masked Language Models (MLMs) (Devlin et al., 2019; Liu et al., 2019; Lan et al., 2020) can also achieve competitive performance using prompt tuning. In practice, MLMs can implement prompt tuning in the form of cloze-style tasks (Devlin et al., 2019; Liu et al., 2019).

With MLMs, prompt tuning has been applied to a large variety of tasks such as factual probing (Perez et al., 2021), text classification (Gao et al., 2021a; Hambardzumyan et al., 2021), relation extraction (Chen et al., 2022b), commonsense reasoning (Ettinger, 2020) and question answering (Khashabi et al., 2020; Jiang et al., 2021), etc.

## 3 Preliminaries of Prompt Tuning

Formally, a text classification dataset can be denoted as $\mathcal{D} = \{\mathcal{X}, \mathcal{Y}\}$, where $\mathcal{X}$ is the text set and $\mathcal{Y}$ is the class set. For each instance $x \in \mathcal{X}$, it is made up of several words $x = \{w_1, w_2, \ldots, w_{|x|}\}$, and is annotated with a label $y \in \mathcal{Y}$.

To bridge the gap between pre-training tasks and downstream tasks (Schick et al., 2020; Gu et al., 2022), prompt tuning is proposed as a cloze-style task to tune MLMs. Prompt tuning consists of a template $T$, a verbalizer $\phi_{\mathcal{Y}}(\cdot)$ and a MLM $\mathcal{M}$. The template (Petroni et al., 2019; Liu et al., 2022) is a textual string with two slots: a *text slot* [T] for text $x$ and an *answer slot* [A] (a <MASK> token) for the cloze-style prediction. The verbalizer is an injective mapping function $\phi_{\mathcal{Y}} : \mathcal{Y} \to \mathcal{V}_{\mathcal{Y}}$ that bridges the class set $\mathcal{Y}$ and the label word set $\mathcal{V}_{\mathcal{Y}}$. Specifically, when the text $x$ is injected into the *text slot*, we get the prompt $T_x$. Then, we can formalize the label probability by feeding the prompt $T_x$ into MLM $\mathcal{M}$ as:

$$p(y|x) = p_{\mathcal{M}}(\text{<MASK>} = \phi_{\mathcal{Y}}(y)|T_x)$$
$$= \frac{\exp(e_{\phi_{\mathcal{Y}}(y)} \cdot h_{\text{<MASK>}})}{\sum_{v \in \mathcal{V}_{\mathcal{Y}}} \exp(e_v \cdot h_{\text{<MASK>}})}, \quad (1)$$

where $h_{\text{<MASK>}}$ is the hidden vector of *answer slot*, and $e_v$ is the embedding of each label word $v \in \mathcal{V}_{\mathcal{Y}}$. Hereafter, we abbreviate <MASK> as <M>. In the training process, we can maximize the learning objective $\sum_{x \in \mathcal{X}} \log p(\text{<M>} = \phi_{\mathcal{Y}}(y)|T_x)$ to tuning MLM $\mathcal{M}$.

## 4 Chain-of-Thought Tuning

We propose Chain-of-Thought Tuning (CoTT), a two-step reasoning framework for Masked Language Models (MLMs) to think step by step, as shown in Fig. 2. The core of CoTT is intermediate steps in natural language form. MLM first generates the intermediate step in step I (Section 4.2), and then uses the generated intermediate step to predict the final result in step II (Section 4.3). We design *convertible slot* [C], a new type of slot

in templates, to introduce intermediate steps flexibly in both steps (Section 4.1). Finally, we fuse the information from both steps to rectify MLM's prediction (Section 4.4).

### 4.1 Convertible Slot in Template

In Section 3, the traditional template contains two types of slots ([T] and [A]) for injecting texts and generating predictions, respectively. However, this template cannot flexibly introduce intermediate steps in natural language form. To overcome this problem, we propose a new type of slot — *convertible slot* [C], which can be converted between injecting and generating intermediate steps based on scenarios.

Specifically, we can incorporate the *convertible slot* [C] at the appropriate position of the template based on semantics. Here, we notate $\mathcal{I}$ as the intermediate step set and prepare a verbalizer $\phi_{\mathcal{I}} : \mathcal{I} \to \mathcal{V}_{\mathcal{I}}$ to establish a connection between $\mathcal{I}$ and intermediate step word set $\mathcal{V}_{\mathcal{I}}$. When the intermediate step is unknown, MLM needs to generate the intermediate step. In this case, we simply fill in [C] with <M>, allowing MLM to make a cloze-style prediction about $\mathcal{V}_{\mathcal{I}}$ to get the intermediate step. [C] is analogous to [A]. When a specific intermediate step $I \in \mathcal{I}$ is given, MLM needs to be provided with such information through the prompt. In this case, we can directly fill the intermediate step word $v_I = \phi_{\mathcal{I}}(I)$ into [C] to inject the appropriate information. [C] is analogous to [T]. In brief, the flexibility of *convertible slot* [C] to convert between [T] and [A] allows MLM to combine intermediate steps to make predictions.

### 4.2 Step I: Generate Intermediate Step

The purpose of step I is to generate the intermediate step by MLM. Specifically, as shown in Fig. 2(a), we fill the text $x$ into [T] and fill an additional <M> into [C] to get the prompt. We still notate such prompt as $T_x$. Then, the probability of $\mathcal{I}$ can be obtained by feeding $T_x$ to MLM $\mathcal{M}$ as:

$$p(I|x) = p_{\mathcal{M}}([\text{C}] = \phi_{\mathcal{I}}(I)|T_x). \quad (2)$$

Eq. 2 implements the first step of CoT: generate the intermediate step based on the text ($x \to I$). Here, we denote MLM's prediction of the intermediate step in Eq. 2 as $\hat{I}$.

**Predict Label in Parallel**

Due to the introduction of the convertible slot, the generation of intermediate steps and labels can be

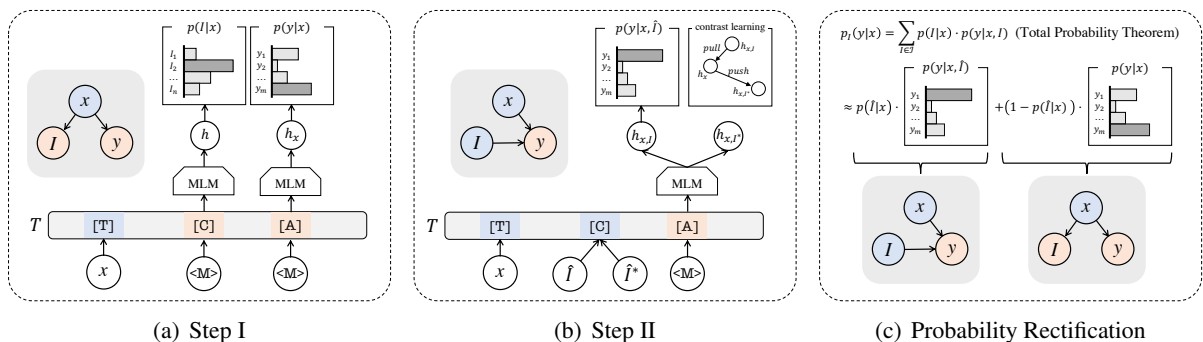

(a) Step I          (b) Step II         (c) Probability Rectification

Figure 2: Overview of Chain-of-Thought Tuning (CoTT). CoTT is a two-step reasoning framework: generate the intermediate step (step I) and use the intermediate step (step II). Then, probability rectification is proposed to rectify MLM's prediction based on the information from both steps. Here, $T$ denotes the prompt, available information is drawn in blue, and generated information is drawn in orange.

simultaneous (multiple <M>s in prompt). Therefore, using the same prompt $T_x$, MLM can predict the label in parallel as:

$$p(y|x) = p_{\mathcal{M}}([\text{A}] = \phi_{\mathcal{Y}}(y)|T_x). \qquad (3)$$

We denote MLM's label prediction in Eq. 3 as $\hat{y}_x$, which is independent of $\hat{I}$. It is worth noting that Eq. 3 does not follow the generation process of CoT: MLM predicts the label in one step without any intermediate steps ($x \rightarrow y$). Due to the lack of a step-by-step reasoning process, we consider $\hat{y}_x$ to be an intuitive prediction.

### 4.3 Step II: Use Intermediate Step

In step II, since the intermediate step $\hat{I}$ is available, MLM can predict the label using both the text and the intermediate step, which is consistent with the second step of CoT ($x \rightarrow y \leftarrow I$).

As shown in Fig. 2(b), we inject $x$ and $\hat{I}$ into the proper slots of the template to obtain the prompt, and we notate such prompt as $T_{x,I}$. Similar to Eq. 1, the label probability with the intermediate step can be expressed as:

$$p(y|x, \hat{I}) = p_{\mathcal{M}}([\text{A}] = \phi_{\mathcal{Y}}(y)|T_{x,I}). \qquad (4)$$

We denote MLM's label prediction in Eq. 4 as $\hat{y}_{x,I}$. Compared to Eq. 3, MLM can perceive and combine the text and the intermediate step in Eq. 4, which makes sophisticated reasoning process possible. Therefore, $\hat{y}_{x,I}$ is considered to be a rational prediction.

**Counterfactual-based Contrastive Learning**

The core of Step II is that MLM needs to integrate the information from both the text $x$ and the

intermediate step $\hat{I}$ to perform logical reasoning. However, this process of information integration lacks explicit guidance. Therefore, we propose counterfactual-based contrastive learning in step II, to guide MLM's information integration by contrasting the hidden vectors obtained from the factual/counterfactual intermediate step.

In most cases, contrastive learning (Gao et al., 2021b) requires an anchor as well as positive and negative samples for each instance. In CoTT, it is natural to consider the hidden vector $h_x$ of [A] in step I as the anchor, and the hidden vector $h_{x,I}$ of [A] in step II as the positive sample. To construct a negative sample, we sample the counterfactual intermediate step $\hat{I}^*$ based on the probability distribution of intermediate steps in Eq. 2 as:

$$\hat{I}^* \sim p_{\notin \hat{I}}(I|x) = \begin{cases} \frac{p(I|x)}{1 - p(\hat{I}|x)} & I \neq \hat{I}, \\ 0 & I = \hat{I}, \end{cases} \qquad (5)$$

where $p_{\notin \hat{I}}(I|x)$ refers to the normalized probability after masking the prediction of the intermediate step $\hat{I}$. Then, similar to step II, we inject counterfactual intermediate step $\hat{I}^*$ as well as the text $x$ to the template to obtain the counterfactual prompt $T_{x,I^*}$, and feed $T_{x,I^*}$ to MLM to get the hidden vector $h_{x,I^*}$ of [A] as the negative sample.

Following (Chen et al., 2020a), we design a small neural network projection head $g(\cdot)$ that maps each hidden vector into the projection space. In this study, we employ a MLP with one hidden layer to obtain the projection vector $z$ as:

$$z = g(h) = W^{(2)} \cdot \sigma(W^{(1)} \cdot h), \qquad (6)$$

where $\sigma$ is a ReLU nonlinearity, $W^{(1)}$ and $W^{(2)}$ are the learnable parameters. We take the cosine similarity of projection vectors as the similarity function because it implicitly normalizes each vector. The similarity between two projection vectors $z_i$ and $z_j$ can be described as:

$$\mathrm{sim}(z_i, z_j) = \frac{z_i^\top \cdot z_j}{\|z_i\| \cdot \|z_j\|}. \tag{7}$$

Then, the counterfactual-based contrastive loss $\mathcal{L}_c$ is defined as:

$$\mathcal{L}_c = -\log \frac{e^{\mathrm{sim}(z_x, z_{x,I})/\tau}}{e^{\mathrm{sim}(z_x, z_{x,I})/\tau} + e^{\mathrm{sim}(z_x, z_{x,I^*})/\tau}}, \tag{8}$$

where $\tau$ denotes the temperature parameter.

By contrasting the similarity of $h_x$ and $\{h_{x,I}, h_{x,I^*}\}$ in the projection space, MLM learns to distinguish whether $x$ and $I$ match: when they do, the hidden vectors of two steps are similar, and vice versa. Overall, counterfactual-based contrastive learning forces MLM to perceive the relationship between texts and intermediate steps more fine-grained, and integrates the perceived information into the hidden vector of [A].

### 4.4 Probability Rectification

In the two-step process mentioned in Section 4.2 & 4.3, we combine $x$ and $\hat{I}$ to obtain the label probability $p(y|x, \hat{I})$. However, this is actually an estimation of the label probability. From the perspective of the total probability theorem, with the consideration of intermediate steps, the exact label probability should be expressed as:

$$p_I(y|x) = \sum_{I \in \mathcal{I}} p(I|x) \cdot p(y|x, I). \tag{9}$$

In step II, $p_I(y|x)$ in Eq. 9 is estimated to be equal to $p(y|x, \hat{I})$ in Eq. 4. The meaning of this estimation is that even if $I \neq \hat{I}$, $p(y|x, I) = p(y|x, \hat{I})$ still holds true. There is a drawback to this estimation: large estimation errors will occur when MLM is ambiguous about intermediate steps ($p(\hat{I}|x)$ is relatively low), which is known as exposure bias (Yang et al., 2018).

However, it is computationally costly to strictly follow Eq. 9 to calculate the exact label probability, as it requires MLM to repeat $|\mathcal{I}|$ times for step II. Therefore, our proposed probability rectification method aims to simplify Eq. 9 by efficiently

estimating the label probability utilizing the information from two steps. Specifically, we assume that when $I \neq \hat{I}$, we have:

$$D_{KL}(p(y|x, I)\|p(y|x)) < D_{KL}(p(y|x, I)\|p(y|x, \hat{I})), \tag{10}$$

where $D_{KL}$ refers to the Kullback-Leibler Divergence. The assumption implies that, compared to using an inconsistent intermediate step, the estimation without using intermediate steps is relatively more accurate. This is consistent with the human perception of intermediate steps. Therefore, we replace $p(y|x, I)$ in Eq. 9 with $p(y|x)$ for all cases satisfying $I \neq \hat{I}$, then the label probability can be estimated more exactly as:

$$
\begin{aligned}
p_I(y|x) &= p(\hat{I}|x) \cdot p(y|x, \hat{I}) + \sum_{I \neq \hat{I}} p(I|x) \cdot p(y|x, I) \\
&\approx p(\hat{I}|x) \cdot p(y|x, \hat{I}) + (1 - p(\hat{I}|x)) \cdot p(y|x).
\end{aligned} \tag{11}
$$

Eq. 11 is the rectified label probability. Essentially, the probability rectification is an adaptive weighted probability based on the probability of the intermediate step $p(\hat{I}|x)$, as shown in Fig. 2(c): when $p(\hat{I}|x)$ is high, we consider that $\hat{I}$ is more trustworthy, so $p_I(y|x)$ is closer to the prediction in step II $p(y|x, \hat{I})$, and vice versa, $p_I(y|x)$ is closer to the prediction in step I $p(y|x)$. Probability rectification is efficient and does not introduce additional computational complexity, but rather judiciously integrates known information from both steps.

### 4.5 Training Details

During the training process, each prediction made in two steps should be optimized, therefore, the loss of CoTT consists of three prediction losses as well as a contrastive loss $\mathcal{L}_c$. We employ the Cross-Entropy loss to calculate the prediction loss. The loss of CoTT $\mathcal{L}$ is denoted as:

$$
\mathcal{L} = \frac{1}{|\mathcal{X}|} \sum_{x \in \mathcal{X}} \underbrace{-\alpha \cdot \log p(I|x) - \log p(y|x)}_{\text{Step I}} \\
+ \underbrace{\beta \cdot \mathcal{L}_c - \log p(y|x, \hat{I})}_{\text{Step II}}, \tag{12}
$$

where $\alpha, \beta$ are the weights to balance each loss respectively, and $|\mathcal{X}|$ represents the number of instances in the dataset.

## 5 Experiments

### 5.1 Datasets and Evaluation Metrics

To verify the effectiveness of CoTT, we conducted extensive experiments on two traditional NLU

| Dataset | # train | # dev | # test | # label |
|---------|---------|-------|--------|---------|
| **WOS** | 30070 | 7518 | 9397 | 141 |
| **TACRED** | 68124 | 22631 | 15509 | 42 |
| **TACREV** | 68124 | 22631 | 15509 | 42 |
| **ReTACRED** | 58465 | 19584 | 13418 | 40 |

Table 1: The statistic details of the four datasets, where # represents the number of instances in each set.

tasks: Hierarchical Classification (HC) and Relation Extraction (RE). For HC task, we conducted our experiments on Web-of-Science (WOS) dataset (Kowsari et al., 2017). WOS contains abstracts of published papers from *Web of Science*, and the labels of WOS have two hierarchies: 7 domains and 134 areas. We treated the domain label as the intermediate step of the reasoning process. We measured the results with Micro-$F_1$ and Macro-$F_1$. For RE task, we experimented on three relation classification datasets: TACRED (Zhang et al., 2017), TACREV (Alt et al., 2020), ReTACRED (Stoica et al., 2021). TACRED was a large-scale sentence-level relation extraction dataset, which was obtained via crowd-sourcing. TACREV corrected the errors in TACRED, and ReTACRED addressed some shortcomings of TACRED, refactoring its training set, development set and test set. ReTACRED also modified a few relation types. For all these datasets, we considered the NER types of subject and object as the intermediate step (refer to Han et al. (2021)), and we adopted $F_1$ score as the metric for evaluation. The statistic details of datasets are illustrated in Table 1.

## 5.2 Baselines

For comparison with LLMs in NLU tasks, we employed two state-of-the-art LLMs in both tasks: davinci-text-003 and gpt-3.5-turbo via the OpenAI API. For HC task, the focus of the recent methods is to exploit label semantics: HiAGM (Devlin et al., 2019) introduced hierarchy-aware structure encoders for modeling label dependencies; HTCInfoMax (Deng et al., 2021) improved HiAGM by text-label mutual information maximization and label prior matching; HiMatch (Chen et al., 2021) matched the text semantics and the label semantics in a joint embedding space; HGCLR (Wang et al., 2022a) directly embedded the hierarchy into a text encoder; HPT (Wang et al., 2022b) handled this task from a multi-label MLM perspective based on prompt tuning. To better compare the performance of CoTT, we also compared our

method with two vanilla prompt tuning methods: HardPrompt and SoftPrompt[1]. For RE task, as fine tuning of MLMs achieved promising results, we fine tuned two traditional MLMs: BERT (Devlin et al., 2019) and Roberta (Liu et al., 2019), as well as two knowledge-enhanced MLMs: SpanBERT (Joshi et al., 2020) and KnowBERT (Peters et al., 2019) as baselines. Since CoTT is based on prompt tuning, we also employed two prompt tuning methods HardPrompt and PTR (Han et al., 2021) as baselines. Experimental details of LLMs can be found in Appendix A.

## 5.3 Implementation Details

Following the setting of the previous study, we adopted bert-base-uncased and roberta-base in Hugging Face library (Wolf et al., 2019) as the base architecture for HC task and RE task, respectively. We set the batch size to 8, and used Adam (Kingma and Ba, 2015) as the optimizer, the learning rate was initially 1e-5 and decayed by 0.5 every 2 epochs. The weight decay was set to 1e-2. After training models for 10 epochs, we selected the best checkpoint on the training set for evaluation. For weighting coefficients, we set $\alpha = 0.1$, $\beta = 0.1$. We set $\tau = 1.0$. The manual templates we designed for HC task and RE task are as follows:

```
HC: [T], the domain is [C], the area is [A].
RE: [T], the SUBJ [C] is [A] of the OBJ [C].
```

where SUBJ and OBJ refer to the subject and the object of each instance, respectively. Since it was a challenge to design appropriate label words to distinguish different labels in the verbalizer (Han et al., 2021), following (Chen et al., 2022a; Wang et al., 2022b), we created the learnable virtual label word $v_y$ for each label $y \in \mathcal{Y}$.

## 6 Results and Analysis

### 6.1 Main Results

Table 2 & 3 exhibit the experimental results of CoTT as well as other compared baselines on HC and RE. It can be observed that our proposed CoTT achieved state-of-the-art results on all datasets. This reflected the superiority of CoTT in NLU tasks. In addition, we found that the performance of LLMs was much worse than MLMs on both tasks, which demonstrated that LLMs still cannot master some NLU tasks well.

---

[1]The concepts of HardPrompt and SoftPrompt originate from Wang et al. (2022b).

| Methods | Micro-$F_1$ | Macro-$F_1$ |
|---|---|---|
| **with LLMs** | | |
| text-003 | 40.50 | 22.25 |
| ChatGPT | 48.50 | 31.83 |
| **Fine tuning MLMs** | | |
| Vanilla Fine Tuning | 85.63 | 79.07 |
| HiAGM | 86.04 | 80.19 |
| HTCInfoMax | 86.30 | 79.97 |
| HiMatch | 86.70 | 81.06 |
| HGCLR | 87.11 | 81.20 |
| **Prompt tuning MLMs** | | |
| HardPrompt | 86.39 | 80.43 |
| SoftPrompt | 86.57 | 80.75 |
| HPT | 87.16 | 81.93 |
| CoTT | **87.46** | **82.49** |

Table 2: Experiment results of different methods on WOS dataset. text-003 and ChatGPT refer to `davinci-text-003` and `gpt-3.5-turbo`, respectively. The best performance would be **bold**.

## Results on HC

As shown in Table 2, CoTT outperformed all baselines on WOS in both metrics. Compared with vanilla fine tuning, CoTT achieved 1.83% and 3.42% improvement on Micro-$F_1$ and Macro–$F_1$, respectively. Besides, vanilla HardPrompt and SoftPrompt can exceed many fine tuning baselines, which revealed that the superiority of prompt tuning remains even in small-scale MLMs. On the basis of prompt tuning, CoTT further improved the performance of MLM, which shows the effectiveness of introducing intermediate steps.

## Results on RE

Table 3 shows that our CoTT continued to perform better than other baselines on three RE datasets. Specifically, CoTT exceeded fine tuning Roberta by 2.9%, 3.6% and 2.7% on $F_1$ score on TACRED, TACREV and ReTACRED, respectively. Note that although no extra data were introduced, CoTT could achieve better performance than those knowledge-enhanced MLMs with fine tuning. This demonstrated that even if task-specific information was already contained in knowledge-enhanced MLMs, fine tuning struggled to take full use of it (Wang et al., 2022b).

## 6.2 Ablation Study

We conducted an ablation study to investigate the independent effect of each module in CoTT, and the experimental results are illustrated in Table 4.

| Methods | E.D. | TACRED | TACREV | ReTACRED |
|---|---|---|---|---|
| **with LLMs** | | | | |
| text-003 | w/o | 48.1 | 51.0 | 55.3 |
| ChatGPT | w/o | 44.9 | 43.9 | 52.1 |
| **Fine tuning MLMs** | | | | |
| BERT | w/o | 68.4 | 77.2 | 87.7 |
| Roberta | w/o | 68.9 | 77.0 | 87.3 |
| SpanBERT | w/ | 70.8 | 78.0 | 85.3 |
| KnowBERT | w/ | 71.5 | 79.3 | 89.1 |
| **Prompt tuning MLMs** | | | | |
| HardPrompt | w/o | 70.1 | 79.5 | 89.0 |
| PTR | w/o | 70.8 | 80.2 | 89.1 |
| CoTT | w/o | **71.8** | **80.6** | **90.0** |

Table 3: Experiment results of different methods on TACRED, TACREV and ReTACRED. 'E.D.' refers to 'Extra Data', 'w/o' means that no additional data is used for pre-training and fine-tuning, and 'w/' means that extra data are used for data augmentation.

| Models | Micro-$F_1$ | Macro-$F_1$ |
|---|---|---|
| Vanilla Prompt Tuning | 86.39 | 80.43 |
| CoTT | **87.46** | **82.49** |
| \ PR | 87.37 | 82.24 |
| \ CCL | 87.30 | 82.28 |
| \ PR&CCL | 87.17 | 82.08 |

Table 4: Ablation results of CoTT on WOS dataset. PR and CCL refer to Probability Rectification and Counterfactual-based Contrastive Learning. \ denotes the removing operation.

The basis of CoTT is the introduction of intermediate steps. On this basis, we also proposed Probability Rectification (PR) and Counterfactual-based Contrastive Learning (CCL) to improve the performance of MLM. When we introduced only intermediate steps (\PR&CCL), the performance of MLM clearly improved. Micro-$F_1$ and Macro-$F_1$ were improved by 0.78% and 1.65%, respectively, compared to vanilla prompt tuning. This implied the importance of intermediate steps in the reasoning process. As we added PR and CCL, the performance of MLM continued to improve, which demonstrated the effectiveness of both modules.

## 6.3 Case Study

To concretely demonstrate the performance of CoTT using intermediate steps, we selected several cases from WOS dataset for evaluation, as shown in Table 5. Here, we recorded only the top three labels by probability in each prediction. The complete texts and the full names of labels can be found in Appendix B.

| Text | Domain | Area | | |
|---|---|---|---|---|
| | $p(I\|x)$ | $p(y\|x)$ | $p(y\|x,\hat{I})$ | $p_I(y\|x)$ |
| ... a native european rodent species, suffered a significant contraction in its geographical ... | Bio. (0.87) Civ. (0.06) Psy. (0.03) | Gene. (0.83) RS (0.02) PCR (0.02) | Gene. (0.86) MB (0.03) PCR (0.02) | Gene. (0.85) MB (0.03) PCR (0.02) |
| tuberculosis is one of the most common infectious diseases in china, while delayed patient ... | Med. (0.52) Bio. (0.43) CS (0.03) | PCR (0.64) Gene. (0.07) H/A (0.04) | H/A (0.30) FI (0.12) PCR (0.04) | PCR (0.33) H/A (0.17) FI (0.08) |
| ... inhabits mangroves and estuarine shores in the west pacific. ... ribosomal RNA (12s) genes ... | Bio. (0.53) MAE (0.24) Civ. (0.17) | PCR (0.25) Gene. (0.19) MB (0.06) | Gene. (0.54) PCR (0.19) MB (0.15) | Gene. (0.37) PCR (0.22) MB (0.11) |

Table 5: Three cases of CoTT on WOS dataset. The prediction probability of each result is in parentheses, and the ground truth in each case is drawn in orange.

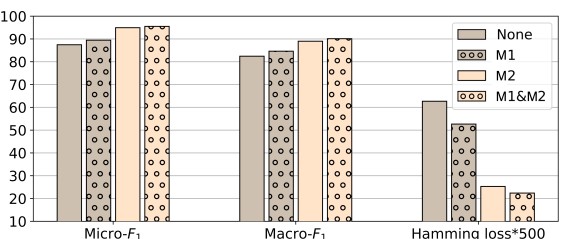

Figure 3: Performance of CoTT in monitoring the reasoning process. M refers to the Monitor.

In Table 5, the first row shows how CoTT would behave under ideal conditions: in step I, MLM's prediction of intermediate steps *Bio.* (0.87) was correct with high confidence, which implied that MLM clearly understood this case, so the label prediction *Gene.* (0.86) in step II was more trustworthy. In this case, probability rectification can be disregarded. However, MLM was sometimes ambiguous about intermediate steps. In the second row, the probabilities of *Med.* (0.52) and *Bio.* (0.43) were close, according to the analysis in Section 4.4, label prediction *H/A* (0.30) can be significantly biased. Therefore, probability rectification is necessary here. In this case, the label prediction was rectified to *PCR* (0.33). The situation in the third row was akin to that in the second row, with the main difference being that the rectified label prediction *Gene.* (0.37) remained unchanged. Essentially, $p(I|x)$, $p(y|x)$, and $p(y|x,\hat{I})$ contain abundant information, while probability rectification can adaptively integrate the information embedded in these probabilities.

## 6.4 Reasoning Process Monitoring

Because of the introduction of intermediate steps, CoTT is no longer an end-to-end method, but can additionally monitor the reasoning process (the distribution of the reasoning process can be found in Appendix C). Therefore, we can detect anomalies from the reasoning process, thus further improving the performance of CoTT.

**Monitor 1: Self-Consistency of Predictions**
In step I and step II of CoTT, MLM should make the label prediction with/without the intermediate step $\hat{I}$, respectively. Ideally, the two predictions (intuitive prediction $\hat{y}_x$ and rational prediction $\hat{y}_{x,I}$) should be self-consistent. Therefore, we consider the inconsistency of label predictions as an anomaly: if the two predictions are contradictory, it

indicates that MLM may have a misunderstanding of the intermediate step $\hat{I}$, making the predictions relatively less reliable.

**Monitor 2: Correctness of Intermediate Steps**
When the true intermediate step is provided, it becomes an option to monitor whether the prediction of intermediate steps in step I is correct. Reasonably, we consider the incorrect prediction of intermediate steps as an anomaly: If MLM cannot predict the correct intermediate step, then the subsequent label prediction is less credible.

Following these two monitors, we evaluated the performance of CoTT in monitoring the reasoning process on WOS dataset. Hamming loss (Tsoumakas and Katakis, 2007) was introduced as an additional metric. As shown in Fig. 3, these monitors (M1 & M2) can determine a more reasonable decision scope for CoTT, which can significantly improve the performance of CoTT. In practice, this ability to monitor the reasoning process can be of great use when faced with risk-sensitive tasks (e.g., medical (Quinn et al., 2020), judicial (Laptev, 2021)), since reliability is even more important in these scenarios.

## 7 Conclusion

In this study, we propose Chain-of-Thought Tuning (CoTT), a two-step reasoning framework based on prompt tuning to implement step-by-step thinking for MLMs on NLU tasks. Specifically, the two-step framework of CoTT enables MLM to implement task decomposition; and CoTT based on prompt tuning, which allows intermediate steps to be used in natural language form. Experiments demonstrated that CoTT achieved state-of-the-art performance on two NLU tasks. In the future, we aim to scale up MLMs to fundamentally improve their language ability. In addition, using LLMs to implement multi-step reasoning in NLU tasks is also a possible research direction.

## Limitations

There are two main limitations in CoTT. Firstly, the application of CoTT is relatively narrow, as CoTT can only handle NLU tasks with intermediate steps. Secondly, compared to LLMs, CoTT struggles to implement interaction and cooperation between intermediate steps and to handle reasoning tasks with more steps.

## Acknowledgments

This work was supported by the Shanghai Municipal Science and Technology Major Project (2021SHZDZX0102), and the Fundamental Research Funds for the Central Universities.

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

## A  Experimental Details of LLMs

We tested LLMs in the 4-shot scenario due to text length limitations, and we sampled 500 texts in each dataset for evaluation. To enable LLMs to implement NLU tasks, we added the task description and label set to the prompt as:

*Based on the abstract of the paper, choose the domain and area of the paper. The optional domains are:* $\{domain_1, domain_2, \ldots, domain_D\}$. *The optional areas are:* $\{area_1, area_2, \ldots, area_A\}$.
*The abstract of the paper is "$x_1$", the domain is "$domain_1$", the area is "$area_1$".*
. . .
*The abstract of the paper is "$x_4$", the domain is "$domain_4$", the area is "$area_4$".*
*The abstract of the paper is "$x$",*

## B  Supplementary Information in Case Study

To demonstrate the performance of CoTT, we selected several cases from WOS dataset[2] to evaluate in case study section. We display the full names of labels, along with the complete texts of the selected cases in Tables 6 & 7.

## C  Decision Distribution of CoTT

With the introduction of intermediate steps, CoTT is no longer an end-to-end approach, so it is feasible to monitor the reasoning process. We counted the reasoning process of CoTT on WOS dataset, and calculated the decision distribution, as shown in Fig. 4.

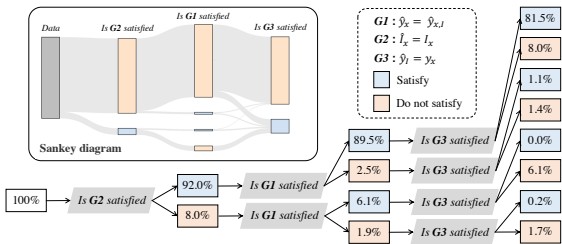

Figure 4: Decision distribution of CoTT

The statistical results show that the decision details provided by the two-phase reasoning framework can help CoTT adaptively adjust the decision scope and thus improve performance. This cannot be achieved by end-to-end methods.

---

[2]It is available at https://huggingface.co/datasets/web_of_science.

---

| # | Text |
|---|------|
| 1 | the garden dormouse eliomys quercinus, a native european rodent species, suffered a significant contraction in its geographical range in the last few decades. the species has disappeared from large parts of central and eastern europe and is considered extinct in some countries. i reviewed the information available on the occurrence and distribution of the species in 26 countries where it was previously reported. present and past introductions outside its native range were also summarised. the garden dormouse is considered extinct in lithuania, finland, and slovakia, probably extinct in belarus, and present with single populations in the netherlands, poland, and slovenia; in slovakia, however, monitoring is necessary to verify recent records. the species is rare and localised in austria, ukraine, romania, and croatia and is in regression in germany, flanders (belgium), czech republic, latvia, and estonia. in 2015, the garden dormouse occupied 49% of its 1978 geographical range and 67% of its 2008 range. south-western europe is the stronghold of the species; it is still common in large parts of portugal, spain, france, and italy. however, there are indications that also in these countries, the species is locally declining. present knowledge cannot explain the extensive regression of the garden dormouse 's range in central and eastern europe. probably, it is the result of the interaction of different factors, acting locally and at a large scale, and related to specific ecological requirements of the species. there is a strong need for research to determine the reasons for the dramatic population and geographical range contraction of the garden dormouse. meanwhile, it is important to monitor this species and to identify appropriate conservation measures. |
| 2 | tuberculosis is one of the most common infectious diseases in china, while delayed patient finding obstructed disease control, especially for smear-negative patients. the current study was undertaken to evaluate the diagnostic accuracy of genexpert mtb/rif compared with conventional methods in the detection of pulmonary tuberculosis patients. a total of 295 spot sputum samples from confirmed pulmonary tuberculosis patients were evaluated from september 2014 to june 2015. each sample was examined by acid-fast bacillus smear microscopy, culture and genexpert mtb/rif. the sputum culture onlowenstein-jensen (l-j)was considered as the gold-standard. after testing by smear, 68.81% (203/ 295) was negative and 31.19% (92/295) was positive. as thegold-standard,l-j culture detected 37.97% (112/295) positive of all specimens, while the positivity for genexpert mtb/rif was 46.44% (137/295). compared with l-j culture, the combined sensitivity, specificity, positive predictive value (ppv) and negative predictive value (npv) for genexpert mtb/rif were 94.64%, 82.97%, 77.37% and 96.18% respectively. for smear-negative specimens, the sensitivity, specificity, ppv and npv for genexpert mtb/ rif were 96.00%, 83.05%, 44.44% and 99.32%; while for smear-positive specimens, the corresponding accuracy values were 94.25%, 80.00%, 98.80% and 44.44%. the findings of study indicated that genexpert mtb/rif assay demonstrated a high sensitivity in detecting mycobacterium tuberculosis compared to smear method and a high npv among smear negative patients. (c) 2017 elsevier ltd. all rights reserved. |
| 3 | this study examined the phylogeography of the barnacle fistulobalanus albicostatus, which inhabits mangroves and estuarine shores in the west pacific. differentiation in the mitochondrial cytochrome c oxidase subunit i (coi) and 12s ribosomal rna (12s) genes of 401 specimens of f.albicostatus was examined in samples from 16 locations in the west pacific, ranging from honshu to southern china. our results revealed that f.albicostatus comprises two major clades exhibiting a coi divergence ranging from 1.25% to 2.8%. clade a demonstrated the widest distribution, ranging from japan to china, and was divided into three subclades occurring in the south china sea (a1), okinawa (a2), and honshu, korea and qingdao (a3). clade b was determined to be endemic to okinawa; i.e. two endemic lineages occur in this island. thus, f.albicostatus resembles several inter-tidal species in having clades that are endemic to okinawan waters. nevertheless, in contrast to the rocky inter-tidal barnacles tetraclita spp. and chthamalus malayensis, f.albicostatus was not found to be separated into continental and oceanic populations, but instead is divided into northern and southern clades, probably because of the yangtze river discharge, which limits gene flow between the northern and southern populations. |

Table 6: The complete texts of the cases in case study.

| Full Name | Abbreviation |
|-----------|--------------|
| Biochemistry | Bio. |
| Genetics | Gene. |
| Civil | Civ. |
| Remote Sensing | RS |
| Molecular Biology | MB |
| Psychology | Psy. |
| Polymerase Chain Reaction | PCR |
| HIV/AIDS | H/A |
| Fungal Infection | FI |
| Computer Science | CS |
| MAE | MAE |

Table 7: The full names of labels in case study.