# OpenReview forum: "Chain-of-Thought Tuning: Masked Language Models can also Think Step By Step in Natural Language Understanding"
_EMNLP/2023/Conference — EMNLP 2023 Main_

### Official Review · Reviewer_bKM7 · 2023-08-04

**Typos Grammar Style And Presentation Improvements:** 1) Line 33
**Soundness:** 4

**Excitement:**

3: Ambivalent: It has merits (e.g., it reports state-of-the-art results, the idea is nice), but there are key weaknesses (e.g., it describes incremental work), and it can significantly benefit from another round of revision. However, I won't object to accepting it if my co-reviewers champion it.

**Missing References:**

NA

**Paper Topic And Main Contributions:**

The paper proposes an approach called "Chain-of-Thought Tuning (CoTT)" that aims to enable MLMs to perform step-by-step thinking in NLU tasks. The CoTT framework involves two steps: Step I, where the MLM generates an intermediate step and label based on the input text, and Step II, where the MLM predicts the final result based on the text and the intermediate step. The paper presents experimental results on hierarchical classification and relation extraction tasks, demonstrating that CoTT outperforms baseline methods.

**Questions For The Authors:**

1) Line 42-48. The paper mentions that LLM needs to be improved for NLU tasks such as GLUE and SuperGLUE NLU datasets. However, the paper only evaluates the proposed approach in two tasks, HTC and RE. Can the paper evaluates the proposed approach in more NLU tasks?

2) While this paper primarily discusses integrating CoT into MLM, it would also be useful to demonstrate the effectiveness of open-source LLMs like LLaMa and ChatGLM using the CoT tuning approach in two steps.

3) The original CoT consists of natural language sentences that make up a reasoning process. However, the proposed CoT uses intermediate labels without detailed explanations. My question is, why are these intermediate labels referred to as CoT?

4) Which dataset did this paper use for the experiment shown in Figure 3?

I will update the final score according to the authors responses.

**Reasons To Accept:**

1) The proposed CoTT technique introduces a two-step reasoning framework based on prompt tuning to facilitate step-by-step thinking in MLMs. The proposed approach is novel.

2) The paper provides a strong motivation for exploring the transferability of CoT from LLMs to MLMs.

3) The paper is well written and easy to follow.

**Reasons To Reject:**

1) The evaluation only covers two NLU tasks: hierarchical classification and relation extraction. Although these tasks are useful for evaluation, a wider evaluation of diverse NLU tasks would strengthen the generalizability of the proposed CoTT framework.

2) Based on the results presented in Tables 2 and 4, the proposed method and its various components do not show significant benefits.

**Reproducibility:**

4: Could mostly reproduce the results, but there may be some variation because of sample variance or minor variations in their interpretation of the protocol or method.

**Reviewer Confidence:**

2: Willing to defend my evaluation, but it is fairly likely that I missed some details, didn't understand some central points, or can't be sure about the novelty of the work.

---

> ### Author Rebuttal · Authors · 2023-08-25
>
> We sincerely thank the reviewer for the efforts and constructive suggestions on our paper. We will further revise our paper based on the reviewer's comments. Here is our response, and we mark the $\color{#660000}{\text{CRITICAL}}$ issues in red that may influence the reviewer's judgments.
>
> _Q1_: Can we evaluate CoTT in more NLU tasks, such as GLUE and SuperGLUE? ($\color{#660000}{\text{CRITICAL}}$)
>
> _A1_: CoTT emphasizes the introduction of intermediate steps to MLMs in NLU tasks, therefore, CoTT imposes some limitations on NLU tasks (__intermediate steps in suitable forms__). We explain this from both task and model perspectives:
> - Task perspective. Not all NLU tasks require intermediate steps. In fact, the process of many NLU tasks in GLUE and SuperGLUE are so simple that introducing intermediate steps is redundant. This is not a situation specific to MLMs, and the original CoT technique is not universal for LLMs, as many text generation tasks do not require intermediate steps (think step by step).
> - Model perspective. MLMs limit the form of intermediate steps. Since MLMs can only perform cloze-style tasks, the intermediate steps cannot be natural language forms of sentences, but only some predefined set of intermediate steps.
>
> We chose HTC and RE to evaluate CoTT also because these two tasks require intermediate steps and intermediate steps can be defined as a finite set. Although CoTT imposes some restrictions on NLU tasks, it is capable of handling much more NLU tasks than HTC and RE, for example, it also has potential in NLU tasks such as information retrieval and common-sense reasoning.
>
> _Q2_: Can CoTT be introduced in open-source LLMs? ($\color{#660000}{\text{CRITICAL}}$)
>
> _A2_: This question is quite important and relates to the motivation of our research -- to improve the ability of MLMs (__not LLMs__) on NLU tasks, so we will explain it in detail.
> - Task perspective: LLMs are not suitable for NLU tasks. Although LLMs have brought the NLP field into a whole new era, essentially the way LLMs solve NLP tasks is to transform everything into text generation tasks. However, many NLP tasks do not suit this transformation, for example, text classification tasks require matching text with a large number of labels, and a more typical task is information retrieval, which requires finding the most relevant content from a huge amount of information. If these NLP tasks are not transformed into text generation tasks and language models are considered only as a textual feature extractors, MLMs (e.g. DeBERTaV3[1]) are still a better choice than LLMs (e.g. LLaMa, ChatGLM).
> - Model perspective: LLMs do not require a two-step process. LLMs are usually autoregressive language models, i.e., they predict the next word based on the previous text. Therefore, LLMs do not need a two-step process like CoTT because autoregressive models automatically use the generated intermediate steps in the generation process to predict the final result. But MLMs predict all <MASK> in parallel, not autoregressively. This is the reason why we introduced a two-step process: we want MLMs to also use the intermediate steps for predicting the final result as LLMs do, so in the first step MLMs predict the intermediate steps and then fill the templates with the intermediate steps for the second step. As for the reason why we force MLMs to do what it is not good at, you can refer to our explanation from the task perspective (LLMs cannot solve all NLP tasks).
> - Cost perspective. LLMs cost more. As academic research, cost does not seem to be the most critical reason. However, it has to be said that the cost of training and use of LLMs far exceeds that of MLMs. so, currently most fine-tuned LLMs are also so-called domain LLMs -- i.e., have the ability to solve a range of tasks in a specific domain. Whereas fine-tuned MLMs are much less costly and can therefore be adapted for a specific single task. So, I think a comparison between prompting LLMs and fine-tuning MLMs is a much more reasonable choice.
>
> _Q3_: Why intermediate labels referred to as CoT?
>
> _A3_: The motivation for CoT is to express the reasoning process explicitly[2]. First, do intermediate labels express the reasoning process explicitly? We believe it does, because the intermediate labels we chose for both HTC and RE are a necessary component of the reasoning process, even humans (implicitly) make use of these intermediate labels. Second, why can we only use the form of intermediate labels and not the form of natural language? As stated in _A1_, since MLMs can only perform cloze-style tasks, generating natural language is not something that MLMs are good at. Overall, intermediate labels are a form that fits the reasoning process of MLMs.
>
> _Q4_: Which dataset did we use for the experiment shown in Figure 3?
>
> _A4_: We thank the reviewer for pointing out the oversight in our manuscript. The experimental results presented in Fig. 3 were obtained on the WOS dataset, and we will add this information in subsequent versions of the manuscript.
>
>
>
> [1] He, Pengcheng et al. “DeBERTaV3: Improving DeBERTa using ELECTRA-Style Pre-Training with Gradient-Disentangled Embedding Sharing.” ICLR 2023.
>
> [2] Wei, Jason et al. “Chain of Thought Prompting Elicits Reasoning in Large Language Models.” NeurIPS 2022.

---

### Official Review · Reviewer_AVcC · 2023-08-05

**Soundness:** 3

**Excitement:**

4: Strong: This paper deepens the understanding of some phenomenon or lowers the barriers to an existing research direction.

**Paper Topic And Main Contributions:**

The authors propose Chain-of-Thought Tuning framework in an attempt to unify the benefits of Masked Language Modeling (MLM) and Chain-of-thought (CoT) in two NLU tasks: hierarchical classification and relation classification. Specifically, the framework is divided into 2 steps (1) Intermediate Step Generation, (2) Target Outcome Prediction via Intermediate Step generated in step (1). Additional Counterfactual-based Contrastive Learning (CCL) and Probability Rectification (PR) are also proposed to effectively integrate intermediate steps into the overall framework.

**Questions For The Authors:**

A. What is the performance of different CoT variants [1,2,3] of LLMs for the two evaluated tasks?

B. Why is SoftPrompt not included in Table 3?

C. The interpretation of Figure 3 seems incomplete. For instance, how to determine reasonable decision scope for CoTT based on Figure 3 (Line 592-594)?

D.For CCL, what is the size of the counterfactual step I* (Line 304)? Is there only a single negative sample for CCL? Since CL benefits from larger batch sizes and more negative samples, it would make more sense to enlarge I* for CCL.

[1] Wei et al., Chain-of-Thought Prompting Elicits Reasoning in Large Language Models. NeurIPS 2022.

[2] Wang et al., Self-Consistency Improves Chain of Thought Reasoning in Language Models. ICLR 2023.

[3] Fu et al., Complexity-Based Prompting for Multi-step Reasoning. ICLR 2023.

**Reasons To Accept:**

1. The overall structure of the work as well as motivations for the integration of CoT and individual components is  well written.

2. The adaptation of CoT to NLU tasks is quite interesting and provides fresh perspectives for NLU tasks.

**Reasons To Reject:**

1. The reported performance gain of the proposed framework is marginal when compared to the improvements introduced by simple Prompt Tuning approaches. For instance,for Table 3, out of 2.7% gain over Roberta backbone on ReTACRED, prompting tuning (i.e. HardPrompt) already achieves the gain of 1.7%.

2. The scope of the study is under-specified. It seems that the work focuses on injecting CoT- based approach to small-scale Language Models. If that is not the case, additional relevant CoT baselines for in-context learning of Large Language Models (for text-003 and ChatGPT) are missing in Table 2 and 3 (See Question A).

3. The major components of the proposed frameworks are CCL and PR. Both of them are incremental over the previous methods with minor adaptation for CoT-based prompting proposal.

**Reproducibility:**

4: Could mostly reproduce the results, but there may be some variation because of sample variance or minor variations in their interpretation of the protocol or method.

**Reviewer Confidence:**

3: Pretty sure, but there's a chance I missed something. Although I have a good feel for this area in general, I did not carefully check the paper's details, e.g., the math, experimental design, or novelty.

**Typos Grammar Style And Presentation Improvements:**

Line 135:  the final chosen answers

Line 519 and 524: , -> .

Tense consistency across the manuscript is recommended.

---

> ### Author Rebuttal · Authors · 2023-08-26
>
> We sincerely thank the reviewer for the efforts and constructive suggestions on our paper. We will further revise our paper based on the reviewer's comments. Here is our response, and we mark the $\color{#660000}{\text{CRITICAL}}$ issues in red that may influence the reviewer's judgments.
>
> _Q1_: CoTT has marginal performance gains? (Reasons To Reject 1)
>
> _A1_: Firstly, we consider that a performance gain of around 1% is also acceptable on traditional NLU tasks like HTC and RE. Second, we value the insights that the research provides to the NLP community more than the performance gains, and in this research, the insight we want to provide is to migrate the CoT technique used for LLMs to MLMs.
>
> _Q2_: The major components of CoTT are CCL and PR. (Reasons To Reject 3) ($\color{#660000}{\text{CRITICAL}}$)
>
> _A2_: The core of CoTT is a __two-stage reasoning framework__. Our motivation for proposing CoTT is to introduce the idea of CoT into MLMs so that MLMs can predict the final result more directly using intermediate steps. __Both CCL and PR are auxiliary modules of CoTT__ rather than the main components. From the ablation experiments in Table 4, we can also see that the two-stage reasoning framework of CoTT is significantly better than the baseline even without the inclusion of the two modules, CCL and PR.
>
> _Q3_: The scope of the study is under-specified. More CoT-based baselines are needed. (Question A) ($\color{#660000}{\text{CRITICAL}}$)
>
> _A3_: This question is very important and relates to the motivation of our research, so we will explain it in detail.
> - First, we emphasize the motivation of our research. Currently, LLMs have brought the field of NLP into a whole new era, but essentially the way LLMs solve NLP tasks is by transforming everything into a text generation task. However, NLU tasks (which require matching text with a large number of labels) are not suitable for this transformation. So, given that MLMs are more suitable for solving NLU tasks, we would like to transfer the CoT technique from LLMs to MLMs.
> - The scope of our research is to transfer the CoT technique originally used in LLMs to MLMs. The essential difference between these two language models is autoregressive/masked, rather than the scale of the model (although the scale of LLMs is usually much larger than that of MLMs). An autoregressive language model predicts the next word based on the previous text, whereas a masked language model predicts all <MASK> in parallel. The solution in our study is to transfer the CoT technique used for autoregressive language models to MLMs using a __two-step reasoning framework__ (in Fig. 2).
> - We did not employ more LLMs+CoT baselines because LLMs are not able to efficiently transform NLU tasks into text generation tasks, which results in LLMs performing much less well than MLMs on NLU tasks, so we do not think that the introduction of more LLMs+CoT baselines would have brought more insight to our research. However, we agree with the reviewer's comments and will include more LLMs+CoT baselines in the subsequent version of our paper, although we believe that the performance of these baselines will be much lower than that of MLMs.
>
> _Q4_: Why is SoftPrompt not included in Table 3? (Question B)
>
> _A4_: In fact, HTC and RE are two different NLU tasks with different baseline approaches. SoftPrompt is not widely regarded as a baseline approach in studies on RE, so we do not include SoftPrompt in Table 3. However, we believe that SoftPrompt can solve the RE task, and we will add the result of SoftPrompt on RE in a subsequent version of the paper.
>
> _Q5_: The interpretation of Fig. 3 seems incomplete. How to determine reasonable decision scope for CoTT? (Question C)
>
> _A5_: We propose two monitoring methods in Section 6.4 to improve the performance of CoTT by determining a reasonable decision scope. The two methods are __Monitor 1: Self-Consistency of Predictions__ (Line 570-580) and __Monitor 2: Correctness of Intermediate Steps__ (Line 581-588). And Fig. 3 shows CoTT's performance under these two monitoring methods. The reviewer may have overlooked that M1 and M2 in Fig. 3 are abbreviations for Monitor 1 and Monitor 2.
>
> _Q6_: For CCL, what is the size of the counterfactual intermediate step $I^*$? (Question D)
>
> _A6_: We sample only one counterfactual intermediate step in CCL. On the one hand, more sampling would bring extra computation, and on the other hand, since the counterfactual intermediate steps are sampled from $p_{\notin \hat{I}}(I|x)$, and this probability distribution is likely to be not very uniform, but distributed over certain confusing intermediate steps (e.g., Biochemistry and Genetics). Our aim in CCL is to help MLMs distinguish between these confusing intermediate steps, so the gain from sampling more counterfactual intermediate steps may not be as significant as in CL.

---

### Official Review · Reviewer_xtiA · 2023-08-10

**Soundness:** 3

**Excitement:**

4: Strong: This paper deepens the understanding of some phenomenon or lowers the barriers to an existing research direction.

**Paper Topic And Main Contributions:**

Natural language understanding (NLU) tasks are still difficult for LLMs to solve, and smaller masked language models (MLMs) still outperform them.
Then, as an analogy to Chain-of-Thought (CoT), which improves the performance of LLMs, this paper introduces a step-by-step reasoning method for MLMs.
Specifically, this paper proposes Chain-of-Thought Tuning (CoTT), a framework that implements two-step reasoning by introducing an intermediate step to cloze-style prompt tuning in MLMs, and decomposes NLU tasks and performs step-by-step reasoning.

CoTT outperforms existing fine-tuning and prompt-tuning methods in two NLU tasks, Hierarchical Classification (HC) and Relation extraction (RE), and achieves state-of-the-art performance.
Furthermore, the introduction of an intermediate step allows the reasoning process of the model to be monitored and could increase the explainability of the model.

**Questions For The Authors:**

- A. You introduced contrastive learning in Step II to explicitly guide the integration of information from input text and intermediate step (ll.288-296), but is it not already explicitly guided by the natural language text parts of the template other than the slot?
- B. I am not convinced that introducing contrastive learning. You explain that the prediction in Step I is an intuitive prediction and the prediction in Step II is a rational prediction. Why learning to make the hidden states for them similar leads to integrating information from the input text and the intermediate step?
- C. CoTT seems to require multiple forward and backward processes in training phase. How much the cost compared to the baseline methods? Is the inference cost about twice as much?
- D. You selected the best checkpoints on the training set for evaluation (ll.465-466). Is it natural to use the training set instead of the dev set for the HC and RE tasks?
- E. Can CoTT be combined with knowledge-enhanced MLMs to better exploit their potential?

**Reasons To Accept:**

- Proposed Chain-of-Thought Tuning (CoTT), an interesting framework allowing MLMs to solve NLU tasks by two-step inference.
- CoTT achieves state-of-the-art performance on two NLU tasks, HC and RE.
- Conducted appropriate ablation studies.
- Introducing an intermediate step may allow us to monitor the inference process of the model and increase the explainability of the model.

**Reasons To Reject:**

- Description of the proposed method (Section 4) is not friendly
    - Concrete examples of prompt template are not provided for explaining the method.
    - Especially, it is difficult to understand the explanation of the convertible slot in Section 4.1 without a concrete example because it is before the explanation of steps I and II.
- I do not understand the motivation to introduce contrastive learning (Question A, B)
- The costs of training and inference are not stated (Question C)
- Only two tasks were used in the evaluation, and it is unclear how effective CoTT is for other NLU tasks, especially tasks in GLUE or SuperGLUE.

**Reproducibility:**

3: Could reproduce the results with some difficulty. The settings of parameters are underspecified or subjectively determined; the training/evaluation data are not widely available.

**Reviewer Confidence:**

3: Pretty sure, but there's a chance I missed something. Although I have a good feel for this area in general, I did not carefully check the paper's details, e.g., the math, experimental design, or novelty.

**Typos Grammar Style And Presentation Improvements:**

- Text in Table 5 is too small

---

> ### Author Rebuttal · Authors · 2023-08-27
>
> We sincerely thank the reviewer for the efforts and constructive suggestions on our paper. We will further revise our paper based on the reviewer's comments. Here is our response, and we mark the $\color{#660000}{\text{CRITICAL}}$ issues in red that may influence the reviewer's judgments.
>
> _Q1_: The descriptions in Section 4 are difficult to understand, especially the explanation of the convertible slot. (Reasons To Reject 1)
>
> _A1_: While writing this paper, we also found this problem - how to logically and smoothly introduce *convertible slot* [C] (Section 4.1). Our writing logic is to pre-define *text slot* [T] and *answer slot* [A] in Preliminaries (Section 3). Then in Section 4.1, [C] is treated as a kind of slot for converting between [T] and [A], and in order to help readers understand [C] more conveniently, the notation and expressions in Section 4.1 are referred to Section 3. We do not provide concrete examples in Section 4.1 because we consider the examples to be task-based, and we describe the tasks we use to evaluate CoTT (HC and RE) in Section 5, so the corresponding templates are also organized in Section 5 (lines 469-471).
>
> _Q2_: Why does CoTT introduce contrastive learning to explicitly integrate text and intermediate steps? This integration should be implemented via prompt. (Question A) ($\color{#660000}{\text{CRITICAL}}$)
>
> _A2_: The motivation for introducing contrastive learning is to __prevent MLMs to learn the correlation shortcut__. Due to space constraints, we only briefly introduce the role of contrast learning in the paper. Since the reviewer has some questions about our motivation for introducing contrast learning, we will explain it in detail.
> - What is the correlation shortcut. A correlation shortcut [1,2] is that a deep learning model may make decisions based on the statistical correlation of partial information, even if it is provided with complete information for causal inference. The essence of this phenomenon is that the optimization process of deep learning is unable to distinguish between correlation and causality of data [3]. A common example of this is that a model distinguishes between a cow and a camel based on the context (grass/desert) rather than the object (cow/camel).
> - Why does the correlation shortcut exist in CoTT? Because there is often a strong statistical correlation between intermediate steps and the final results. For example, there is clearly a high statistical correlation between the intermediate step (Biochemistry) and the final result (Genetics). However, it is unreasonable to predict Genetics solely based on Biochemistry (ignoring the text), even though it may seem effective. However, due to the inability to distinguish between correlation and causality, MLMs are likely to learn this correlation shortcut while ignore the correct reasoning process (integrating text and intermediate steps to predict the final result).
> - Why prompt cannot block the correlation shortcut. The main role of prompt in CoTT is to provide the complete information required for causal inference to MLMs through slots. Specifically, prompt makes text and intermediate steps available to MLMs in the most straightforward manner. But as mentioned before, even provided with complete information, MLMs may make decisions based on the statistical correlation of partial information (the correlation shortcut). Therefore, it is necessary to explicitly guide MLMs to integrate text and intermediate steps. (__for Question A__)
> - Why contrastive learning can block the correlation shortcut. To explicitly integrate text and intermediate steps, we introduce Counterfactual-based Contrastive Learning (CCL). Specifically, there are three hidden vectors in CCL: hidden vector $h_x$ without intermediate steps (in step I), hidden vector $h_{x, I}$ with correct intermediate step (in step II), hidden vector $ h_{x, I^*}$ with counterfactual intermediate step (in step II). The goal of CCL is that $h_{x, I}$ is close to $h_x$, and $h_{x, I^*}$ is far away from $h_x$. Intuitively, by contrasting the similarity of $h_x$ and {$h_{x, I}, h_{x, I^*} $}, MLMs have to distinguish whether $x$ and $I$ match: when they do, the hidden vectors of two steps are similar, and vice versa. Overall, CCL forces MLMs to perceive the relationship between text and intermediate steps more fine-grained, and integrates the perceived information into the hidden vector $h$, thereby restricting MLMs from exploiting the correlation shortcut (text cannot be ignored). (__for Question B__)
>
> _Q3_: Why make hidden vectors similar in contrastive learning to integrate information from text and intermediate step? (Question B) ($\color{#660000}{\text{CRITICAL}}$)
>
> _A3_: We explain in detail the reasons for introducing contrastive learning in _A2_. Our interpretation of this question is in the last point in _A2_.
>
> _Q4_: CoTT seems to require multiple forward and backward processes in training phase. How much the cost compared to the baseline methods? Is the inference cost about twice as much?
>
> _A4_: Yes, CoTT requires two forward and backward processes in the training phase and two forward processes in the inference phase. Compared to the most basic baseline (Vanilla Prompt Tuning), the computational effort is about twice as high. However, some of the advanced baselines we have chosen also significantly increase the computational effort. In addition, CoTT allows for more information to be obtained from the two forward processes, which can increase the interpretability of CoTT (Table 5), as well as detecting anomalies to determine a reasonable decision scope (Section 6.4). Overall, CoTT trades increased computational effort for better performance, as well as increased interpretability and reliability.
>
> _Q5_: Is it natural to use the training set instead of the dev set to select the best checkpoints? (Question D)
>
> _A5_: We agree with the reviewer that using dev set to select checkpoints is a more rigorous approach according to the principles of machine learning. However, we did find many studies on HC and RE tasks that used the TRAINING SET to select checkpoints. After our verification, in most cases the two results are consistent. However, we consider that we should follow the principles of machine learning and adopt a more rigorous approach, and we will make corresponding changes in the subsequent versions of the paper.
>
> _Q6_: Can CoTT be combined with knowledge-enhanced MLMs to better exploit their potential? (Question E)
>
> _A6_: We believe that this potential combination is feasible because the proposed CoTT is theoretically applicable to any MLMs. Our study only used traditional MLMs (BERT, Roberta) in order to allow for clearer comparisons with the baselines. We consider the reviewer's comments to be constructive, and we will include the experimental results of CoTT combined with knowledge-enhanced MLMs in the subsequent version of the paper.
>
> [1] Shah, Harshay, et al. "The pitfalls of simplicity bias in neural networks." NeurIPS (2020).
>
> [2] Feder, Amir, et al. "Causalm: Causal model explanation through counterfactual language models." Computational Linguistics (2021).
>
> [3] Fan, Caoyun, et al. "Accurate use of label dependency in multi-label text classification through the lens of causality." Applied Intelligence (2023).

---

### Official Review · Reviewer_VXXs · 2023-08-12

**Soundness:** 4

**Excitement:**

3: Ambivalent: It has merits (e.g., it reports state-of-the-art results, the idea is nice), but there are key weaknesses (e.g., it describes incremental work), and it can significantly benefit from another round of revision. However, I won't object to accepting it if my co-reviewers champion it.

**Paper Topic And Main Contributions:**

This work proposes a chain-of-thought tuning on mask language models to solve the natural language understanding tasks in a step-by-step manner. Inspired by the success of the large language model, decomposing the complex task into multiple intermediate subtasks, the authors design a step-by-step tuning on the mask language model to seek achievement on traditional natural language understanding tasks. The proposed framework consists of a two-step tuning process, where the task is sequentially divided into two stages, with the final prediction being conditioned on the intermediate output. The proposed model shows impressive new SOTA results on hierarchical classification and relation extraction tasks.

**Questions For The Authors:**

1.	The chain-of-though style technique requires the task to be solved in a step-by-step manner, stressing the order of subparts after decomposition. But solution flows are not only a flat chain of thoughts but also others like tree-like style or complex combination.

2.	Since this work mentions the task decomposition and degrade the importance of P(y|x,I) in Line370 - 372, there is a lack of comparison between CoTT with a parallel multi-task framework. As we know, there are many works with multi-task frameworks targeting intermediate output for further better final prediction.

**Reasons To Accept:**

1.	This work shows a comprehensive understanding of task background and challenges.

2.	CoTT is a straightforward but novel solution to study traditional natural language understanding tasks that can be decomposed into two sequential sub-tasks.

3.	This work provides a clear and solid introduction to the model framework,  a rigorous experimental setup, and a plentiful and noteworthy discussion.

**Reasons To Reject:**

1.	The chain-of-though style technique requires the task to be solved in a step-by-step manner, stressing the order of subparts after decomposition. But solution flows are not only a flat chain of thoughts but also others like tree-like style or complex combination.

2.	Since this work mentions the task decomposition and degrade the importance of P(y|x,I) in Line370 - 372, there is a lack of comparison between CoTT with a parallel multi-task framework. As we know, there are many works with multi-task frameworks targeting intermediate output for further better final prediction.

**Reproducibility:**

4: Could mostly reproduce the results, but there may be some variation because of sample variance or minor variations in their interpretation of the protocol or method.

**Reviewer Confidence:**

4: Quite sure. I tried to check the important points carefully. It's unlikely, though conceivable, that I missed something that should affect my ratings.

**Typos Grammar Style And Presentation Improvements:**

For Section 5, you may use simple present tense rather than past tense, because we usually use simple present tense to introduce facts.

---

> ### Author Rebuttal · Authors · 2023-08-26
>
> We sincerely thank the reviewer for the efforts and constructive suggestions on our paper. We will further revise our paper based on the reviewer's comments. Here is our response, and we mark the $\color{#660000}{\text{CRITICAL}}$ issues in red that may influence the reviewer's judgments.
>
> _Q1_: Solution flows are not only a chain but also others like tree-like style or complex combination. ($\color{#660000}{\text{CRITICAL}}$)
>
> _A1_: I agree with the reviewer that a generic solution flow is not in the form of a chain but is much more complex. There are several reasons why our research (CoTT) uses a chain-based solution flow:
> - Language perspective. The motivation for the original CoT is to explicitly represent the reasoning process through language. And language is linear and seems to be best suited for a chain-based solution flow. For LLMs, CoT can perform tree-like solution flow (e.g., complex reasoning, theorem proving), which we attribute to the stronger reasoning ability of LLMs. MLMs do not have the reasoning ability of LLMs, and so we restrict the solution flow to a chain.
> - Flow perspective. Theoretically, any solution flow can be regarded as a combination of chains. So, we consider it is also important to study the basic unit (chain) of the solution flow, especially when current techniques are not sufficient to solve the generic solution flow.
> - Effectiveness Perspective. Our proposed CoTT significantly improves the ability of MLMs in some NLU scenarios. In line with the EMNLP philosophy, we believe that empirical methods (even with limitations) are valuable to the NLP community.
>
> _Q2_: Lack of baselines with a parallel multi-task framework.
>
> _A2_: We chose the baselines based on previous researches. For the HC task, the intermediate labels themselves are included in the evaluation metrics, so the intermediate labels are included in the training objectives. Therefore, some baselines of HC (Vanilla Fine Tuning, HardPrompt, SoftPrompt) can all be considered as parallel multi-task frameworks. For the RE task, the intermediate labels are independent of the evaluation metrics. This may have led previous researches to ignore the baselines of parallel multi-task frameworks in RE, so we do not include this type of baselines. We consider the reviewer's comment of parallel multi-task framework to be constructive, and we will include baselines for parallel multi-task frameworks in the RE task in subsequent versions of the paper.
>
> Additionally, we would like to express our gratitude to the reviewer for pointing out the presentation problems in our paper. This feedback is valuable not only for this paper but also for our future research.

---

### Meta-Review · Area_Chair_k91h · 2023-09-25

**Recommendation:** 4

**Metareview:**

This journal paper presents a novel framework called Chain-of-Thought Tuning (CoTT) for solving natural language understanding (NLU) tasks.
CoTT is designed to decompose NLU tasks into two sequential sub-tasks and incorporates prompt tuning to enable step-by-step reasoning in masked language models (MLMs). It also includes ablation studies and highlights the potential for increased explainability of the model by introducing an intermediate step. The integration of CoTT to NLU tasks is well-motivated and provides fresh perspectives for the field.
However, the significant components of the proposed frameworks are minor adaptations from previous methods. The evaluation only covers two types of tasks and uncovers how effective the chain-of-thought style technique is for other NLU tasks.
Moreover, the reported performance gain of the proposed framework is marginal compared to simple Prompt Tuning approaches.
Therefore, a more comprehensive evaluation of diverse NLU tasks would strengthen the proposed framework's generalizability.

Overall, the paper is well-written, easy to follow, and demonstrates the transferability of CoT concepts from language models to masked language models.

---

### Decision · Program_Chairs · 2023-10-07

**Decision:**

Accept-Main

**Comment:**

This journal paper presents a novel framework called Chain-of-Thought Tuning (CoTT) for solving natural language understanding (NLU) tasks.
CoTT is designed to decompose NLU tasks into two sequential sub-tasks and incorporates prompt tuning to enable step-by-step reasoning in masked language models (MLMs). It also includes ablation studies and highlights the potential for increased explainability of the model by introducing an intermediate step. The integration of CoTT to NLU tasks is well-motivated and provides fresh perspectives for the field.
However, the significant components of the proposed frameworks are minor adaptations from previous methods. The evaluation only covers two types of tasks and uncovers how effective the chain-of-thought style technique is for other NLU tasks.
Moreover, the reported performance gain of the proposed framework is marginal compared to simple Prompt Tuning approaches.
Therefore, a more comprehensive evaluation of diverse NLU tasks would strengthen the proposed framework's generalizability.

Overall, the paper is well-written, easy to follow, and demonstrates the transferability of CoT concepts from language models to masked language models.